# EEG based Emotion Recognition of Image Stimuli

## Abstract

Emotion is playing a great role in our daily lives. The necessity and importance of an automatic Emotion recognition system is getting increased. Traditional approaches of emotion recognition are based on facial images, measurements of heart rates, blood pressure, temperatures, tones of voice/speech, etc. However, these features can potentially be changed to fake features. So to detect hidden and real features that is not controlled by the person are data measured from brain signals. There are various ways of measuring brain waves: EEG, MEG, FMRI, etc. On the bases of cost effectiveness and performance trade-offs, EEG is chosen for emotion recognition in this work. The main aim of this study is to detect emotion based on EEG signal analysis recorded from brain in response to visual stimuli. The approaches used were the selected visual stimuli were presented to 11 healthy target subjects and EEG signal were recorded in controlled situation to minimize artefacts (muscle or/and eye movements). The signals were filtered and type of frequency band was computed and detected. The proposed method predicts an emotion type (positive/negative) in response to the presented stimuli. Finally, the performance of the proposed approach was tested. The average accuracy of machine learning algorithms (i.e. J48, Bayes Net, Adaboost and Random Forest) are 78.86, 74.76, 77.82 and 82.46 respectively. In this study, we also applied EEG applications in the context of neuro-marketing. The results empirically demonstrated detection of the favourite colour preference of customers in response to the logo colour of an organization or Service.

keywords: Electroencephalography (EEG), Brain computer interface (BCI), machine learning, emotion recognition, image stimuli, neuromarketing

## 1 Introduction

Emotion is playing a great role in our daily lives. The necessity and importance of an automatic Emotion recognition system is getting increased. Traditional approaches of emotion recognition are based on facial images, measurements of heart rates, blood pressure, temperatures, tones of voice/speech, etc. However, these features can potentially be changed to fake features. Thus, to detect hidden and real features that is not controlled by the person are data measured from brain signals. There are various ways of measuring brain waves: EEG, MEG, FMRI, etc. On the bases of cost effectiveness and performance trade-offs, EEG is chosen for emotion recognition in this study. The main aim of this study is to detect emotion based on EEG signal analysis recorded from brain in response to visual stimuli. The approaches used were the selected visual stimuli were presented to 11 healthy target subjects and EEG signal were recorded in controlled situation to minimize artefacts (muscle or/and eye movements). The signals were filtered and type of frequency band was computed and detected. Brain computer interface(BCI) based Emotion recognition are used in a variety of applications include advertisement, patient treatment, depression management, music player, human computer interaction, detecting children learning disabilities, assist disabilities with communication, game playing, automatic addition of emotional pictures during conversation,

Submitted to 33rd Conference on Neural Information Processing Systems (NeurIPS 2019). Do not distribute.

emotion enabled avatar, neuromarketing, etc.[1]. To introduce few facts of the human brain, our brain is one of the largest and complex organs of human body. It is the center of consciousness which enables the human to think, innovate, learn and create that makes human different from other animals. It is quite challenging to understand how the brain functioning as it is made from million of million neuron cells (around 100 billion nerves) which in turn communicate trillions of connections (called synapses). This research focus on the outermost layer of human brain which is the cerebral cortex (cerebrum). The cerebrum is broadly divided in to left and right hemispheres, which are symmetrically nearly equal. Each hemisphere is in turn divided into four lobes including Frontal lobe, Parietal lobe, Temporal lobe and Occipital lobes. These lobes get their names from the bones of the skull that overlie them. Human uses peripheral device such as mouse, keyboards, monitor, etc to interact with the computer whereas brain computer interface (BCI) is a device that allows the computer to read the human brain neuro-physiological activity and processes to perform a particular task without using traditional peripheral devices. The typical components of a BCI includes: signal acquisition, pre-processing, feature extraction and pattern recognitions. Signal acquisition, where the brain activity is recorded, pre-processing, where filtering, dimensionality reduction and feature extraction is carried out, pattern recognition where the selected features are used for detecting the target concept in the application. Finally, Post processing could be performed to instruct a particular device/system. The user might receive feedback from the device/system. For example, human can instruct the computer to write what he wants based on just sending thought signals from the brain to the computer. In this research, we use EEG as it is more cost effective with reasonable quality trade-off than other types of neuroimaging approaches. The other reason is that EEG has capability to handle high temporal resolution and can directly measure the brain activity (non-invasive) with simple and portable device [2]. The brainwave activity is broadly divided into five frequency bands. The boundary between the frequency bands is not strict but not varying much. The frequency bands include delta(0.5-4Hz), theta(5-8Hz), alpha(9-12Hz), beta(13-30Hz) and gamma(above 30Hz) [3]. For this study, EEG data is collected using Emotiv EPOC device with 14 electrodes located at AF3, F7, F3, FC5, T7, P7, O1, O2, P8, T8, FC6, F4, F8, AF4. The electrodes are placed according to a 10-20 placement system with sampling rate of 128Hz [4].

## 1.1 Problems:

According to the literature, even though it is possible to measure emotion from EEG signals recorded from stimulated brain in practice, the outputs of BCI related research works are quite different with same stimuli and with brain response of same or different subjects [5]. The other problem is that parts of the brain that responds to emotion is not clearly identified or mixed up research results. For example, emotion is responded either or both on Frontal lobe or temporal lobe. Besides this, the brain wave contains emotion is not clearly known in that whether Alpha frequency band or gamma frequency band. These are some of the problems to motivates us to work on it.

## 1.2 Research questions:

This study attempts to find out answers for the following research questions: (1) What regions of the brain are associated with visual emotion? (2) Which frequency bands of the brain waves are used for emotion recognition? (3) How accurately the chosen features were recognizing emotions using machine learning approaches? Due to space limitation on this report, we tried to present methods and results for answering some of this research questions.

## 2 Related Works

This section briefly presents a few key related works. Researchers reported that there is high correlation between the two hemispheres (i.e. left and right) of the brain in relation to emotional activity. The left brain responds to positive emotion (i.e. joy or happiness) where as the right brain responds to negative emotion (i.e. fear or disgust). The main cause of emotion is the change of alpha power in asymmetry between hemispheres of the brain. In asymmetrical frontal lobe, beta or alpha band from (pre)frontal and parietal asymmetry and gamma band from temporal asymmetry responsible for valence where as prefrontal asymmetry in alpha band and temporal asymmetry in gamma band is also responsible for arousal [6]. In other words, the EEG brain activity from parietal and frontal lobe of the brain is more emotionally informative where as gamma, alpha and beta waves

are more important to discriminate emotional states than other brain wave frequency bands. There are also research gender related emotionality differences in that women are suggested to respond emotional stimuli more than men do [7]. Chauhana et al(2016) [8] developed stress reductions systems based on EEG signal analysis of subjects response to audio or videos. This study tried to filter EEG into the 5 frequencies (alpha, beta, gamma, theta and delta) and applied on real-time emotion recognition of users based on visual and audio stimuli and demonstrated possible real applications.Yang in [9] applied Fishers Linear Discriminant Classifier(FLDA) on TV commercials and the results showed that happiness index of EEG more than behavioral analysis. Hettich et al [10] employed Support Vector machine to classify emotions (pleasant, neutral or unpleasant) caused by a particular auditory stimuli by recording EEG signal. Lin et al in [11] applied support vector machine to classify four emotional states(joy, anger, sad and pleasure) based on EEG responses from music stimuli. The average accuracy of support vector machine is 82.29$\acute{3}$.06. Yisi in [12] developed real-time emotion recognition algorithm based on EEG signal from audio stimuli and identified its possible applications. The approach is success-fully applied as music therapy to help patients to deal with their problems. Jiahui et al [13] investigated subject specific emotion recognition system based on frequency bands of EEG signals from visual stimuli. The online accuracy recognition of this system was 74.17%. Martina et.al in [2] introduced scientific methods of neuromarketing applications based on professional and scientific point of view. It also stated the postulates for applying neuromarketing. On similar study, Bertin et.al in [14] investigated the evaluation of TV commercials whether there is +/- correlation between EEG signals from prefrontal cortex and surveyed based evaluation. The results supports that neural waves supplement the verbal ways of traditional promotion.

# 3 Methods

## 3.1 Data Sets Preparation

We collected and prepared three image data sets for stimuli presentation and classification. These image data sets include: 90 sample images of Geneva Affective Picture Database (GAPED), 8 Colour images and 36 Indian company logo images. As classification algorithms require labeled data sets for building models via in supervised training, we merged class label information for each EEG records of each image stimulus in the data sets.

## 3.2 Hardware and Software Tools

EMOTIV EPOC head sets, Emotiv EPOC TestBench Control panel software and EventIDE are used for EEG brain activity recording. Emotiv headset is relatively simple to setup, it can uniformly capture brain signal from almost all regions of cerebral cortex and it is cost effective. Saline was applied to properly hydrate electrodes and fully contact with the scull. EventIDE was used to record the Power Spectrum Density (PSD) of all 14 channels along with five frequency bands including: theta, alpha, $low\_beta$, $High\_beta$ and gamma frequency bands. Therefore, the total of 70 channels are recorded for a total of 11 subjects(person) samples for each of the three image data sets. These bands are filtered and finally, saved in a file for further processing. The proposed approach in this study has consists of six stages: image stimulus presentation, subjects, EEG Signal recordings, Signal Filtering, feature extraction and classification. For pre-processing and building machine learning models, Weka is used.

# 4 Results and Discussion

To answer research question 1 and 2, the top ranked features for each subject are extracted using Relief algorithm in [15]. The brain frequency bands where it has top ranked features are counted for each subject. On the basis of this result, 37.5% of the subjects are responded to emotional images with alpha brain waves. The brain frequency bands and the channel numbers are counted in each of the three experiments on the three data sets. For this study, we build supervised machine learning models implemented in Weka. These includes Bayesian Network, J48(decision tree), Adaboost(meta learner) and Random forest. After fine tuning the selected machine learning models, it predicts an emotion type (positive/negative) in response to the presented stimuli. Finally, the performance of these models are tested on test sets. The average accuracy of machine learning algorithms (i.e. J48, Bayes Net, Adaboost and Random Forest) are 78.86, 74.76, 77.82 and 82.46 respectively. In conclusion, we tried

to address three key issues. First, we empirically identified the brain regions which more responsible for emotion. On the basis of feature evaluation result, frontal lobe is more emotionally informative than other regions of the brain. Second, alpha and theta frequency bands are more discriminative than other brain frequency waves for emotion recognition. Third, random forest outperformed the other three algorithms (bayes Net, J48 and adaboost) in detecting the customers emotion of image stimuli regardless of domain of application and gender. For real world applications, we have also demonstrated EEG developed machine learning models in the context of neuro-marketing. The results of this research work provides intelligence actions to detect the favourite colour preference of customers in response to the logo colour of an organization or Service as it revealed in the experimental set ups.

## 5 Conclusion

This project is an EEG based Emotion recognition of image stimuli where there are a number of challenges including the variability of emotion recognition system that in turn caused by lack of quality in the recording of EEG data due to the variability among level of attention of subjects, the variability arise in multiple session, the variability caused by muscle movement, the variability due to machine noise, differing physiology of subjects, differing cognitive patterns and differing behavior of subjects [5]. Thus, we tried handle our bests to regulate the causes of variability in EEG data recordings. For example, besides precautions during recordings, we applied filters for removing artifacts. Thus, we recommended interested researchers to work on EEG based researches in the areas of neuromarketing, TV ads evaluation, product branding, product preferences, disability treatment, stress management, just to name a few.

## References

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
