# OpenReview forum: "EEG based Emotion Recognition of Image Stimuli "
_NeurIPS.cc/2019/Workshop/Neuro_AI — Submitted to Real Neurons & Hidden Units @ NeurIPS 2019_

### Official Review · AnonReviewer1 · 2019-09-19
**Validity of the approach to EEG based emotion detection remains unclear**

**Clarity:** 2

**Comment:**

While the collected dataset could yield interesting observations about neural signatures of emotions I remain unconvinced given the present analyses and interpretation. A clearer description of the followed steps and an interpretation grounded in the neuroscientific literature will benefit the paper.

**Category:**

AI->Neuro

**Clarity Comment:**

The paper unfortunately remains quite underspecified and has several typos. The reasoning about brain function remains vague.

**Evaluation:**

1: Very poor

**Importance:**

2: Marginally important

**Importance Comment:**

Emotion detection from neural data could have interesting applications.

**Intersection:**

2: Low

**Intersection Comment:**

Standard application of a suite of ML algorithms to a neural dataset

**Rigor Comment:**

It is very hard to infer from the paper if the analyses have been performed in a rigorous manner. There's no information on exact preprocessing nor a motivation for the use of this collection of ML algorithms.

**Technical Rigor:**

1: Not convincing

---

### Official Review · AnonReviewer2 · 2019-09-27
**Unclear presentation of results**

**Clarity:** 2

**Category:**

AI->Neuro

**Clarity Comment:**

There are very few details given about the methods and results.

**Evaluation:**

2: Poor

**Importance:**

2: Marginally important

**Importance Comment:**

Emotion recognition is a topic of wide interest.

**Intersection:**

2: Low

**Intersection Comment:**

This paper uses machine learning algorithms to classify emotion from EEG.

**Rigor Comment:**

It is very difficult to judge the technical rigor of this paper as the methods and results are barely explained. It is not even clear what the classification task even is exactly. The motivation and intro should be much smaller and much more space should be give to explaining the experiment, the data, the features, the classification task, plots of the results, significance tests, and clear interpretation.

**Technical Rigor:**

1: Not convincing

---

### Decision · Program_Chairs · 2019-10-01

**Decision:**

Reject

**Comment:**

Unfortunately, we had more submissions than we could accept and based on the review process, we have decided not to accept your submission.  Nevertheless, thank you for your submission and interest in our workshop.